# Morpho-Histological Studies of the Gastrointestinal Tract of the Orange-Rumped Agouti (*Dasyprocta leporina* Linnaeus, 1758), with Special Reference to Morphometry and Histometry

**DOI:** 10.3390/ani12192493

**Published:** 2022-09-20

**Authors:** Kegan Romelle Jones, Roger Edmund John, Venkatesan Sundaram

**Affiliations:** Department of Basic Veterinary Sciences (DBVS), School of Veterinary Medicine (SVM), Faculty of Medical Science (FMS), University of the West Indies (UWI), St. Augustine Campus, St. Augustine 999183, Trinidad and Tobago

**Keywords:** agouti, gastrointestinal tract, gross anatomy, histology, histometry, morphometry

## Abstract

**Simple Summary:**

The agouti (*Dasyprocta leporina*) is a neo-tropical rodent that has the potential to be domesticated. These animals are considered to be omnivores from studies conducted in the wild and in captivity. However, an in-depth morphometric and histometric analysis of the gastrointestinal tract of this animal was never conducted. Thus, the objective of this paper was to give a quantitative analysis of the gastrointestinal tract of the agouti and relate it to the feeding habits of the animals. This study showed that the agouti had an esophagus which was lined with keratinized epithelium and esophageal glands. The stomach was simple, with well-developed gastric glands, and the small intestines accounted for the majority of the animal’s digestive tract in length. The long cecum and well-developed colon indicated that this animal had the ability to digest fibrous material in the diet. Based on the analysis of the gastrointestinal tract, these animals can be fed a dry omnivorous diet, with protein sources from either vegetation or animal matter.

**Abstract:**

The morphology of the gastrointestinal tract (GI) is a strong indicator of a species’ dietary habits. The objective of this study was to describe and quantitatively analyze the gross and microanatomy of the digestive tract of the orange/red-rumped agouti (*Dasyprocta leporina*) and relate it to the feeding habits of this animal. The digestive tracts of six adult males were used for this study. The results showed that the esophagus was thick (mean thickness of 1023.78 ± 28.97 μm) and lined by keratinized epithelium with scant esophageal glands. Mucosa-associated lymphocytic infiltration was robust throughout the GI tract. These findings suggest that the esophagus was well adapted to a coarse diet. The simple stomach with well-developed gastric glands in the fundus region (mean thickness of 605.39 ± 28.68 μm) was indicative of an adaptation to a carnivorous diet. The small intestine constituted approximately 80% of the length of the GI tract. The remarkable development of the jejunum with a greater villus length (mean thickness of 182.50 ± 27.38 μm) indicated a greater absorptive capacity in frugivorous and carnivorous diets. The long cecum and well-developed colon clearly indicated that the GI tract was well adapted to frugivorous and herbivorous diets. Overall, *D. leporina* showed that it is well adapted to an omnivorous diet. These results suggest that *D. leporina* can be fed a balanced omnivorous dry diet with a high protein content of plant or animal origin that is well suited to the architecture of the GI tract in captivity.

## 1. Introduction

The vertebrate digestive tract is a highly specialized structure through which food enters the organism and is digested, nutrients are absorbed, and waste products are excreted [1]. The morphology of the digestive tract can influence the efficiency of digestion and is closely related to feeding habits. Ecological conditions and feeding habits lead to changes in the basic metabolic rate, which, in turn, lead to changes in the size of the digestive tract [2,3,4]. Changes in the digestive tract can have a seasonal pattern [5,6,7] and often reflect the physiological state of an organism [8,9].

Orange-rumped agoutis (*Dasyprocta leporina*) are large rodents belonging to the order Caviomporpha, a rodent order with diverse anatomical characteristics among its members [10]. It is a very popular exotic meat in Trinidad and Tobago [11,12]. An estimated 90,000 agoutis are hunted in Trinidad and Tobago each year during the hunting season [13]. These large forays and loss of habitat can lead to threats to the survival of this species. In recent years, a growing number of breeders [12] in Trinidad and Tobago have attempted to domesticate this animal through breeding to meet the demand for meat. The attempts to domesticate and intensively produce *D. leporina* in Trinidad and Tobago have sparked interest in studying the biology of this animal.

In any intensive livestock production system, knowledge of digestive-system anatomy leads to improvements in feeding and nutrition [14]. *D. leporina* feeds omnivorously and consumes both plant and animal matter [15,16]. A preliminary anatomical study on the digestive tract of agouti was conducted by Garcia et al. [17] that lacks a detailed quantitative and qualitative analysis of the microscopic anatomy of the digestive tract. Therefore, the objective of this work was to perform a comprehensive anatomical and histological study, with special emphasis on morphometric and histometric measurements of the gastrointestinal tract of agouti in order to understand the digestive biology of this animal. Further research with various experimental diets would be a better model to analyze the histological microstructure in depth.

## 2. Materials and Methods

Six adult male *D. leporina* were collected for this study from the Neotropical Wildlife Unit (Figure 1) at the University of the West Indies Field Station (UFS) in Valsayn, (10°38′15″ N, 61°25′39″ W) Trinidad and Tobago (Figure 1). The study was carried out in the Anatomy Unit of the School of Veterinary Medicine, The University of The West Indies, St. Augustine Campus, Trinidad and Tobago. The Campus Research Ethics Committee, The University of the West Indies, St. Augustine, Trinidad and Tobago (No. CREC-SA 1095/07/2021) approved all animal experiments in accordance with the National Institutes of Health’s Guide for the Care and Use of Laboratory Animals [18].

Animals were sedated with a solution of xylazine and ketamine hydrochloride at 5 mg/kg body weight and 35 mg/kg body weight, respectively. Live weight was measured and recorded. Animals were then euthanized by intracardiac injection of pentobarbital sodium (release 300 mg/mL). Gross in situ morphological observations were then made and photographed. The entire gastrointestinal tract from the esophagus to the anus was removed, and the length of the different segments was measured. Gross anatomical features were examined in each segment, and specimens were collected and preserved in formalin and embedded in paraffin for histologic examination. The relative size of each segment of the digestive tract was determined by dividing the segment length by the length of the entire digestive tract. Tissue samples collected for histological examination were fixed in 10% buffered neutral formalin and processed for routine paraffin embedding. Sections of 5–8 μm in thickness were cut and stained with routine hematoxylin and eosin (H&E). An Olympus BX51 microscope (Olympus, Tokyo, Japan) with cell Sens imaging software (ver. 1.12) was used to examine histologic features and obtain histometric measurements and photomicrographs.

Histometric measurements were taken by using a micrometric scale for microscopy to measure specimens. The thickness of the mucosa, submucosa, tunica muscularis, and serosa layers was obtained from the esophagus, stomach (cardiac, fundic, and pyloric regions), small intestine (duodenum, jejunum, and ileum), and large intestine (cecum, colon, and rectum). In the small-intestine region, villi length and width and crypt height and width (μm) were measured. The length of the villi was measured from the tip of the villus to the beginning of the crypt. The width of the villi was measured transversely from the widest point of the villi. The depth of the crypt was measured from the bottom of the villi to the base of the crypt. Crypt width is the transverse diameter of the crypt. The average number of goblet cells per 10,000 μm^2^ area was also measured. Ten measurements were taken per slide, and three slides were taken for each segment of the gastrointestinal tract; measurements were averaged to determine the value for each animal. Descriptive statistics were calculated by using IBM SPSS Statistics V21 software (IBM, Armonk, NY, USA).

## 3. Results

### 3.1. Gross Anatomy of the GI Tract

The mean length of the entire GI tract was 780.5 ± 50.62 cm (Table 1). The esophagus had a mean length of 16.41 ± 1.45 cm, and the width of the esophagus remained fairly uniform throughout its length, merging into a simple rounded stomach (Figure 2 and Figure 3). The stomach was unicameral, had an average length of 12.70 ± 0.71 cm, and was thin-walled. Three regions of the stomach were distinguished: the cardia, the fundus, and the pylorus. The cardia was the junction between the esophagus (Figure 4) and the stomach, and the pylorus formed the dilated posterior end of the stomach. The pylorus was thick-walled and had an average diameter of 3.2 ± 0.03 cm. A thick muscular band, the pyloric sphincter, marked the constricted junction between the stomach and the small intestine.

The small intestine (SI) consisted of a simple smooth-walled tube with an average length of 603.9 ± 10.15 cm. The small intestine consisted of 77.4% (Figure 5) of the length of the entire gastrointestinal tract (Figure 3). There were no demarcations that distinguished it from the duodenum, jejunum, and ileum. Overall, however, the small intestine accounted for a large proportion of the total length of the digestive tract (Figure 5; Table 1). The duodenum has a sigmoid flexure in which the pancreas is located. The large intestine occupied the caudal end of the digestive tract and was divided into the cecum, colon, and rectum. The most prominent of the three subdivisions of the large intestine (LI) was the large thin-walled greenish cecum. The cranial and caudal portions of the cecum had mean diameters of 4.2 + 0.04 and 2.4 ± 0.02 cm, respectively, making it only a smaller diameter than the stomach. The outer surface of the cecum had dorsal and ventral longitudinal bands and 14 saccular outpouchings. The average length was 23.55 ± 0.64 cm. The ileum and colon united with the cecum to form the ileocolic junction.

The colon and rectum formed a continuous tube without sharp demarcations. The first segment of the colon was enlarged. After the first segment, the rest of the colon was gradually reduced in diameter until it merged into the rectum. The rostral part of the colon had a slightly tortuous spiral consisting of two folds, namely a centripetal fold and a centrifugal fold. The distal end of the colon was bulging due to fecal matter in the lumen. The mean diameter of the colon at the cranial and caudal ends was 2.4 ± 0.02 and 1.3 ± 0.01 cm, respectively. The rectum had a fairly uniform diameter, averaging 1.3 ± 0.01 cm. The mean length of the colon and rectum was 123.55 ± 24.64 cm.

### 3.2. Histology of the GI Tract

The mucosa of the esophagus was folded and lined by stratified keratinized squamous epithelium with an average thickness of 1023.78 ± 28.97 μm. The lamina propria consisted of lymphocytic infiltration throughout the esophagus. In addition, scattered lymph nodules were also found. The muscularis mucosae was distinct and consisted of smooth muscle fibers. The submucosa was the thickest layer at 465.67 ± 74.78 μm and consisted of a dense irregular connective tissue layer with scant esophageal glands, lymphocytes, blood vessels, and nerves. The tunica muscularis consisted of outer longitudinal and inner circular muscle fibers. In the present study, the cranial tunica muscularis consisted mainly of skeletal muscle and transitioned to smooth muscle at the cardiac sphincter. The cervical esophagus is covered by adventitia, a loose connective tissue layer containing lymphatic and blood vessels and nerves. The thoracic esophagus is covered by serosa, a loose connective tissue layer with an outer mesothelial covering.

The division of the stomach into three distinct regions is based on the length and composition of the gastric glands: cardiac, fundic, and pyloric. The surface stomach of *D. leporina* was essentially lined and folded with simple columnar epithelium. The gastric pits were characterized by the shape and location of the three regions. The lamina propria consisted of a loose connective tissue layer that was rich in capillaries and lymphoid cells and was completely occupied by gastric glands. The tunica muscularis consisted of two layers of smooth muscle, namely inner circular and outer longitudinal layers of smooth muscle fibers (Figure 5). The outer surface of the stomach was covered by the serosa, which consisted of loose connective tissue, blood vessels, lymphatic vessels, and nerves.

The mucosa in the cardiac region was characterized by densely packed tortuous tubular mucous glands that lay beneath the surface of the simple columnar epithelium. The surface was indented into numerous sharp gastric pits that opened freely into the lumen and were wide and deep. Gastric glands made up most of the mucosa below the pits (Figure 6, Figure 7 and Figure 8). The glands were relatively short and had few superficial mucosal cells. Some parietal cells were observed at the base of the glands, and many neck cells were identified. There were also two layers of smooth muscle in the tunica muscularis.

The mucosal folds and gastric pits in the fundus region were shallower in the mucosa, and the mean thickness was measured to be 605.39 ± 28.68 μm, which was higher than in other regions (Table 2). The gastric glands were straight simple tubular glands that ran parallel to each other and extended deep into the mucosa to reach the muscularis mucosae (Figure 8). The surface of the epithelium consisted of fewer mucosal cells. The glands in the fundus were divided into three regions: a basal region, a middle region, and an upper isthmus. The upper isthmus contained both parietal and surface mucosal cells, whereas the parietal and mucosal cells were abundant in the middle region of the glands. The basal region consisted mainly of a few parietal cells and mainly chief cells. There was a much-thickened muscularis mucosae with some striations that extended into the lamina propria between the gastric glands to the gastric folds. The submucosa was composed of dense connective tissue. The tunica muscularis consisted of two layers of smooth muscle, namely the thin outer longitudinal layer and the thick inner circular layer.

The region of the pylorus was similar to the fundus, with the following exceptions: the gastric glands were more branched and shorter, the gastric pits were short, and more principal chief cells and fewer parietal cells were observed. The thick tunica muscularis formed a sphincter at the gastroduodenal junction (Figure 8)

The duodenum was lined by a simple columnar epithelium composed of absorptive cells with enteroendocrine cells and some goblet cells. (Figure 9). Crypts and flattened villi characterized the duodenal mucosa, and the plicae circulares were not prominent. Between the villi were short tubular structures, Lieberkühn crypts. There was extension of the lamina propria between the crypts into the villous core, which had lymphocyte infiltration. The muscularis mucosa was thin, with a large number of duodenal glands beneath. The muscular coat was well developed and consisted of two layers of smooth muscle: a longitudinal outer layer and a circular inner layer.

Finger-like long villi characterized the jejunal mucosa with an average thickness of 182.50 ± 27.38 μm (Table 3), and plicae circulares were observed (Figure 10). The epithelium on the surface had few goblet cells and many absorptive cells. Paneth cells were located at the base of each distinct crypt. Diffuse lymphocyte infiltration was observed in the lamina propria. Two layers of smooth muscle were present in both the tunica muscularis and duodenum. The tunica serosa was typical, and the smooth muscle layers in the tunica muscularis were also thicker than in the duodenum.

Plicae circulares were clearly observed in the ileum and were more prominent than in the duodenum and jejunum (Figure 11). The number of goblet cells was increased numerically with an average of 24.67 ± 2.73 cells (Table 3) compared with the other two regions (duodenum, 2.00 ± 0.63; ileum, 5.50 ± 1.52). In the jejunum, there was similarity with the appearance of the other structures. No Peyer’s patches were observed, but diffuse lymphoid infiltration was noted.

There were no villi in the mucosa of the cecum, and there were very well-developed plicae circulares (Figure 12). There were more goblet cells than simple columnar cells. Intestinal crypts and diffuse lymphoid tissue were observed in the mucosa, whereas no glands were found in the submucosa. The tunica serosa was typical, and there was a very thick tunica muscularis.

In the mucosa of the colon, there was a simple columnar epithelial layer with a large number of goblet cells. (Figure 13). The crypts were straight and tubular with absent villi. Lymphocytic infiltration was observed in the lamina propria. Tubular, straight mucous glands were identified in the mucosa, and there were two muscle layers in the muscular layer.

The rectal mucosa was lined with a large number of goblet cells, and the tubular straight glands extended throughout the thickness of the mucosa (Figure 14, Table 4). A markedly thick circular inner area was observed.

## 4. Discussion

In this study, an attempt was made to explore the anatomy and histology of the GI tract of *D. leporina*. The results of the study show that the anatomy is typical of monogastric fermenters, and the histology resembles the tubular organs of mammals with few variations.

The esophagus is a relatively thick muscular tube with all four histological layers, similar to the findings in *D. leporina* [17]. The epithelium is lined with keratinized epithelium, similar to that found in ruminants, horses, and rodents, an adaptation to roughage feeding, whereas in humans, dogs, and cats, a non-keratinized stratified epithelium is found [19]. This suggests that the epithelial tissue renews itself, which is essential for maintaining the mucosal layer and protecting it from pathogen invasion and mechanical abrasion [19]. A scant number of esophageal glands (not shown) indicated that *D. leporina* can process roughage. The increased number of glands is associated with increased buffering capacity and absorption of volatile fatty acids, as reported in *Lama glama* [20]. The tunica muscularis is composed predominantly of striated muscle, similar to that found in dogs; ruminants; and rodents, including mice, rats, and hamsters [21], and transitions into smooth muscle at the cardiac sphincter.

The overall morphology of the mammalian stomach varies widely among species, although there are some basic structural differences. Stomach morphology is strongly influenced by adaptation, type of food, frequency of food intake, duration and need for food storage, body size and shape, etc. The stomach in the present study was unilocular and simple and approximately pear-shaped. The body and fundus of the stomach are not clearly demarcated, as noted by Garcia et al. [17]. Grossly, the stomach of the agouti is similar to that of the lappe [22], but it differs from the capybara, which has a gastric diverticulum [23]. The capybara and lappe are also hystricomorphic rodents, but the capybara is an herbivore, and the lappe is a frugivore. The intermediate, simple morphology and size of the stomach may indicate that *D. leporina* is better adapted to an omnivorous diet. 

Histologically, the stomach consists of the same structure as in mammals in terms of layers and cell types. The gastric mucosa is lined by a simple columnar epithelium with gastric glands. The mucosa of the fundus and pyloric glands are much better developed than the cardiac region. However, the parietal cells are the predominant cell type; the chief cells were also increased, indicating that the glandular stomachs are well equipped to digest a coarse diet. The presence of a well-developed muscularis mucosa adapts the stomach to distension and has the function of preventing the deformation of the glandular layer. In addition, the muscularis mucosa may be related to the absorption of easily digestible substrates such as disaccharides and short-chain fatty acids [24].

Schieck and Millar [25] compared the morphology of the digestive tract of 35 rodent species and also found that mass and length were greater in herbivores than in granivores, explaining that the length of the small intestine in small mammals does not reflect the amount of dietary fiber in each species. In the present study, the small intestine was found to be the largest segment of the GI tract, and dietary habits in the wild and in captivity showed that it could process herbivorous food very well. However, no significant differences were found in measurements of the small intestine of the white-footed mouse (*Peromyscus leucopus*), an omnivorous species, and the meadow mouse (*Microtus pennsylvanicus*), an herbivorous species, but measurements of the hindgut showed diet-specific anatomical differences [9]. The intestine of *D. leporina* has a simple histological organization compared to that of rodents. The typical histology of the intestinal tube consists of four layers. The innermost layer is the mucosa with the muscularis mucosa, the submucosa, the tunica muscularis, and the tunica serosa, which is the outermost layer [19]. Mucous cells were found in the intestinal mucosa, and there was a numeric increase (non-significant) in number in the ileum. Although the amount and composition of the mucilaginous substance produced by the gastric and intestinal epithelium and cells of the glands might be mainly related to nutrition and the functional importance of the environment in the stomach and intestine of *D. leporina*, the villi are more developed (based on numeric difference) in the jejunum than in the other areas, suggesting that the jejunum is the most active site of absorption.

The crypts in the present study are mostly unbranched, and some branched crypts have more than one base per opening. They are thicker than the muscularis mucosa. Many authors found a close correlation between the depth of the crypts and the proliferation rate of epithelial cells [26,27]. In the present study, the crypts are well developed, suggesting that the absorbing cells of the intestine proliferate very well, which, in turn, contributes to better absorption. Some studies have shown that increased energy requirements lead to an increase in the length and volume of the small intestine [28,29] and that dietary habits are closely related to intestinal morphology and structure [30,31].

In the present study, the lamina propria throughout the small intestine showed strong lymphocyte infiltration. In particular, lymphocytes were numerous in the ileum, which was not observed in normal Peyer’s patches [32]. This suggests that the intestine is well adapted to cope with infectious agents that may be present in the ingested food.The large intestine (LI) was wide and short in length compared to the small intestine in *D. leporina*. The LI was highly developed, suggesting increased adaptation to water absorption from fruits [33,34]. The results suggested that *D. leporina*, like opossums, can consume a large amount of fruit in the wild [35]. The cecum in the present study is large and well-developed, and there is also a large and well-developed colon, suggesting that this animal is a hindgut fermenter and can store and process large amounts of herbivorous food. Overall, *D. leporina* possessed the typical structures of mammalian species in terms of microanatomy. However, as for the SI and LI, they have a considerable amount of lymphoid tissue, indicating that these species have a well-developed immunological response.

## 5. Conclusions

The anatomy and histology of the GI tract of *D. leporina* were studied. The results showed that the well-folded esophagus lined with keratinized epithelium with scant esophageal glands, lymphatic infiltration, and a thick wall (1023.78 ± 28.97 µm) indicated adaptation to a large amount of coarse food. The simple stomach with a well-developed gastric gland suggests a carnivorous diet, and the strongly developed small intestine with the remarkable development of villi in the jejunum suggests that they absorb food very well and saturate it for a long time in the intestine. The long cecum and well-developed colon clearly indicate that the GI tract is adapted to frugivorous and herbivorous diets. Overall, *D. leporina* suggests that it is well adapted to its omnivorous feeding behavior. The morphology and histo-architecture of the various regions of the GIT, including the morphometry and histometry of the alimentary canal, justify these adaptations. These results provide qualitative and quantitative background information on the gross morphology and histology, as well as on some functional components of the GIT in *D. leporina*, which are essential for generating data for this animal. However, further detailed studies at the ultrastructural level are needed to confirm these results. Detailed feeding studies have to be performed to obtain the nutritional requirements for this animal, as well as enzymatic studies.

## Figures and Tables

**Figure 1 animals-12-02493-f001:**
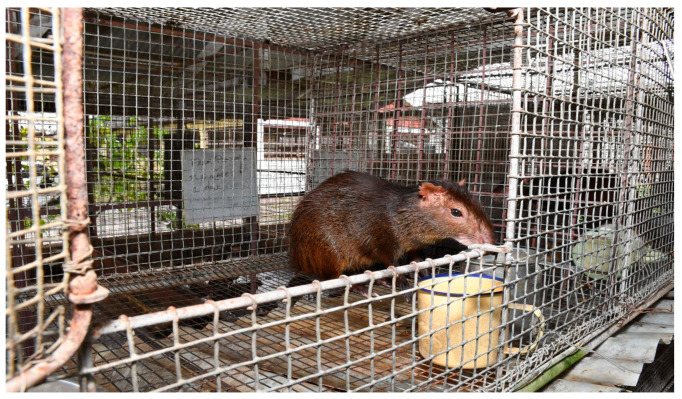
Photograph showing *D. leporina* in a cage housing at the Neotropical Wildlife Unit at University Field station.

**Figure 2 animals-12-02493-f002:**
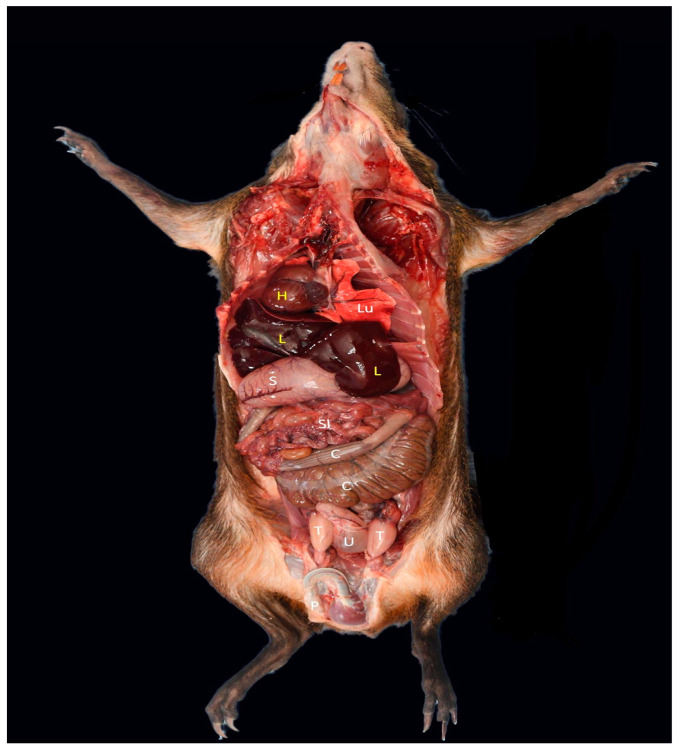
Photograph showing the anatomical view of visceral organs in situ of an adult male *D. leporina*. H—heart; L—liver; Lu—lungs; S—stomach; SI—small intestine; C—colon; T—testis; U—urinary bladder; P—penis.

**Figure 3 animals-12-02493-f003:**
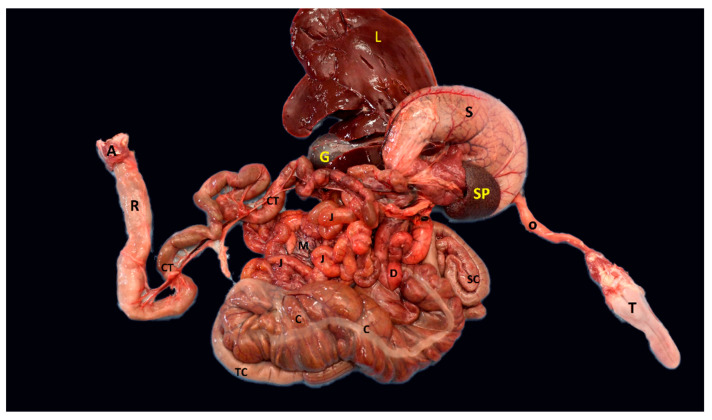
Photograph showing the anatomical view of the gastrointestinal tract of an adult male *D. leporina*. T—tongue; O—esophagus; SP—spleen; L—liver; G—gall bladder; S—stomach; D—duodenum; J—jejunum; C—cecum; SC—spiral colon; TC—transverse colon; CT—terminal colon; R—rectum; M—mesentery; A—anus.

**Figure 4 animals-12-02493-f004:**
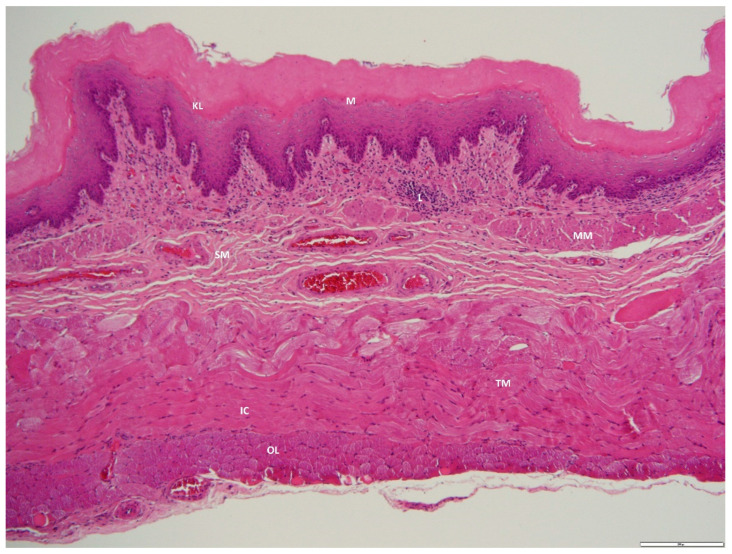
Photomicrograph of a transverse section through the esophagus of an adult male *D. leporina*. H&Ex200. KL—keratinized epithelium; M—mucosa; MM—muscularis mucosa; SM—submucosa; L—lymphocyte infiltration; TM—tunica muscularis; IC—inner circular layer; OL—outer longitudinal layer.

**Figure 5 animals-12-02493-f005:**
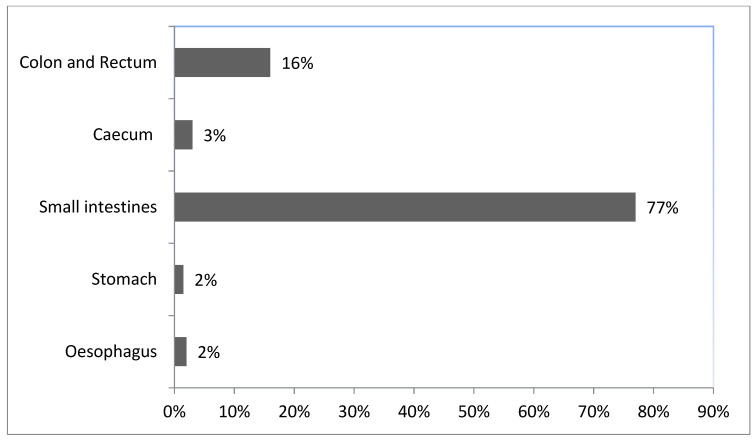
Relative length (%) of the gastrointestinal tract of the agouti.

**Figure 6 animals-12-02493-f006:**
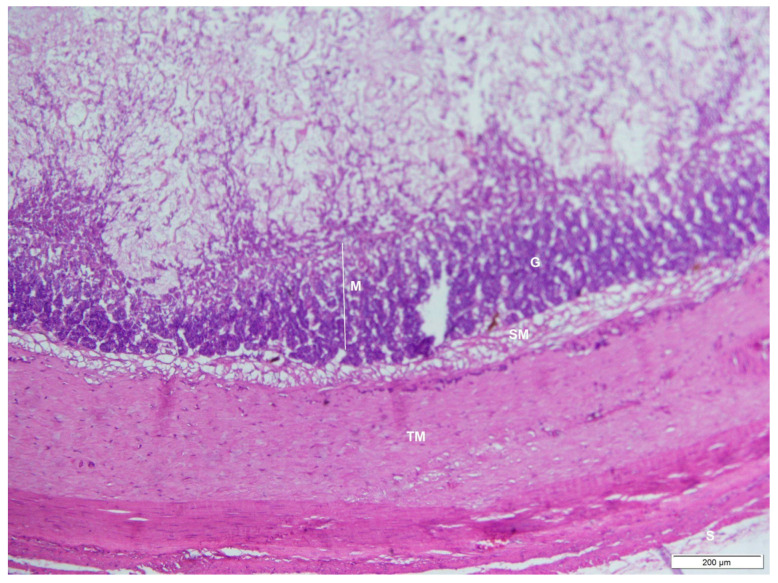
Photomicrograph of a transverse section through the cardiac region of an adult male *D. leporina*. H&Ex200. M—mucosa; G—gastric glands; SM—submucosa; TM—tunica muscularis; S—serosa.

**Figure 7 animals-12-02493-f007:**
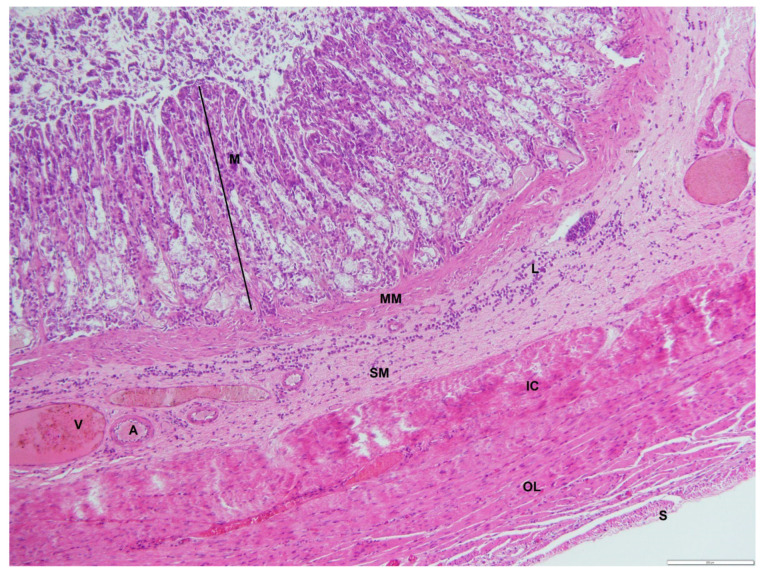
Photomicrograph of a cross-section through the fundic region of an adult male *D. leporina*. H&Ex200. M—mucosa; MM—muscularis mucosa; SM—submucosa; L—lymphocyte infiltration; A—arteriole; V-venule; IC—inner circular layer; OL—outer longitudinal layer; S—Serosa.

**Figure 8 animals-12-02493-f008:**
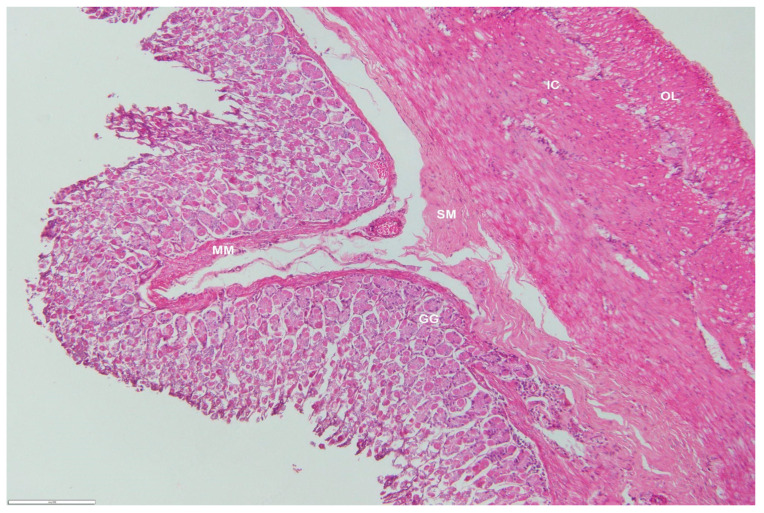
Photomicrograph of a transverse section through the pyloric region of an adult male *D. leporina*. H&Ex200. GG—gastric glands; MM—muscularis mucosa; SM—submucosa; IC—inner circular layer; OL—outer longitudinal layer.

**Figure 9 animals-12-02493-f009:**
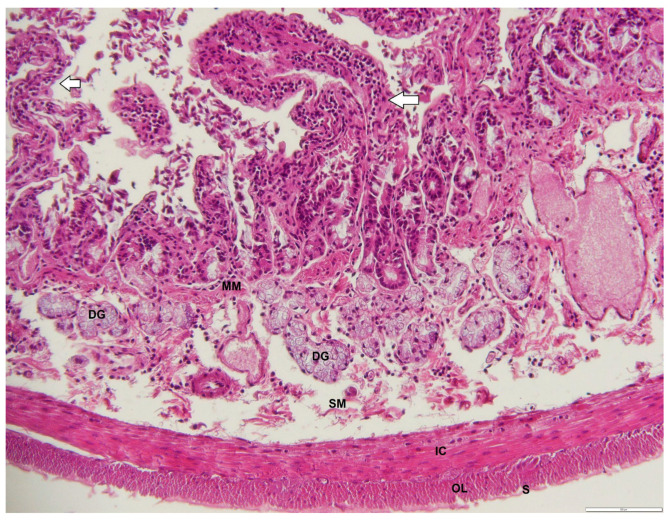
Photomicrograph of a transverse section through the duodenum of an adult male *D. leporina*, showing villi (arrows). H&Ex200. MM—muscularis mucosa; DG—duodenal glands; SM—submucosa; IC—inner circular layer; OL—outer longitudinal layer; S—serosa.

**Figure 10 animals-12-02493-f010:**
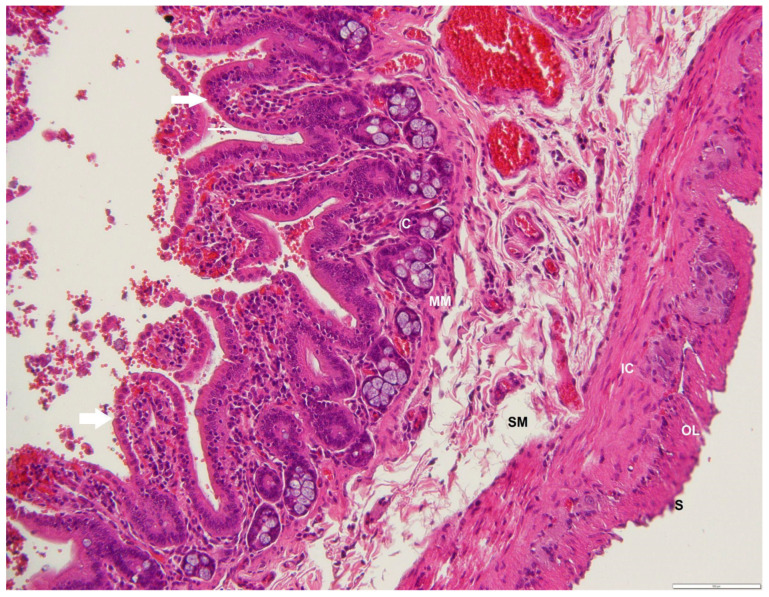
Photomicrograph of a transverse section through the jejunum of an adult male *D. leporina*, showing villi (arrows). H&Ex200. MM—muscularis mucosa; C—intestinal crypts; SM—submucosa; IC—inner circular layer; OL—outer longitudinal layer; S—serosa.

**Figure 11 animals-12-02493-f011:**
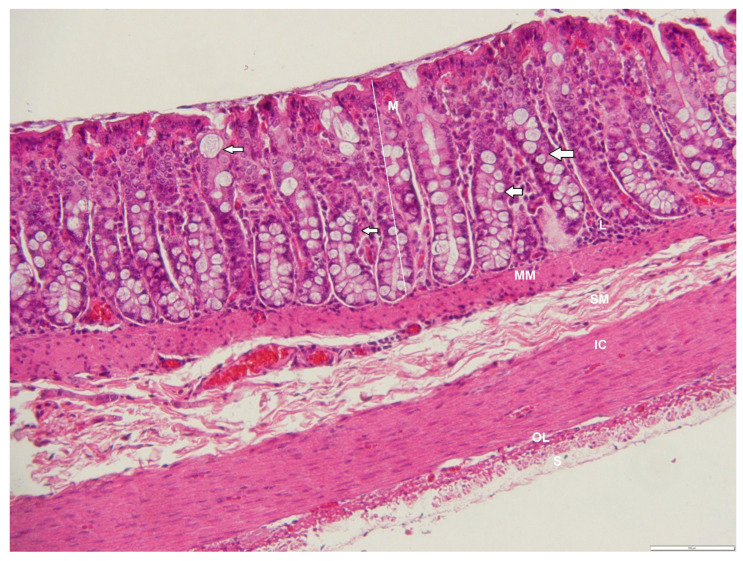
Photomicrograph of a cross-section through the ileum of an adult male *D. leporina*, showing numerous goblet cells in the mucosa (arrows). H&Ex200. M—mucosa; MM—muscularis-mucosa; L—lymphocyte infiltration; SM—submucosa; IC—inner circular layer; OL—outer longitudinal layer; S—serosa.

**Figure 12 animals-12-02493-f012:**
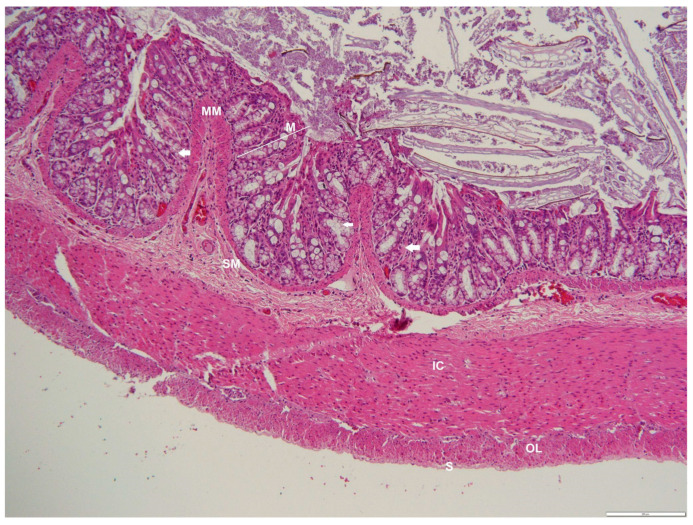
Photomicrograph of a transverse section through the cecum of an adult male *D. leporina*, showing goblet cells (arrows). H&Ex200. M—mucosa; MM—muscularis mucosa; SM—submucosa; IC—inner circular layer; OL—outer longitudinal layer; S—serosa.

**Figure 13 animals-12-02493-f013:**
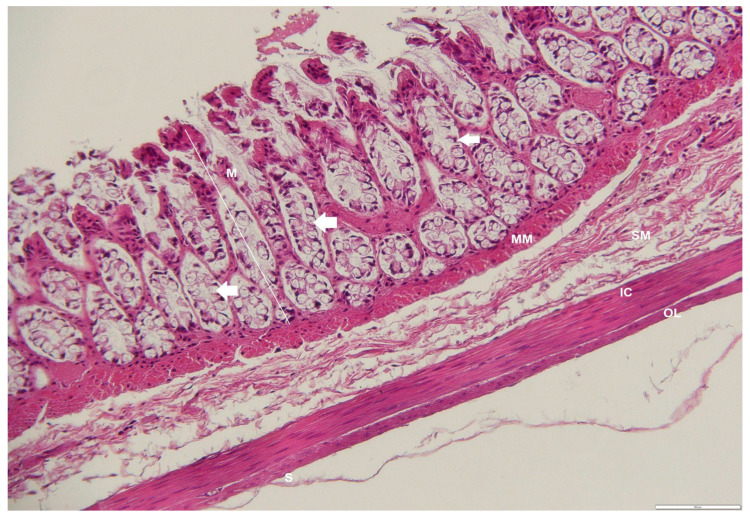
Photomicrograph of a cross-section through the colon of an adult male *D. leporina*, showing goblet cells (arrows). H&Ex200. M—mucosa; MM—muscularis mucosa; SM—submucosa; IC—inner circular layer; OL—outer longitudinal layer; S—serosa.

**Figure 14 animals-12-02493-f014:**
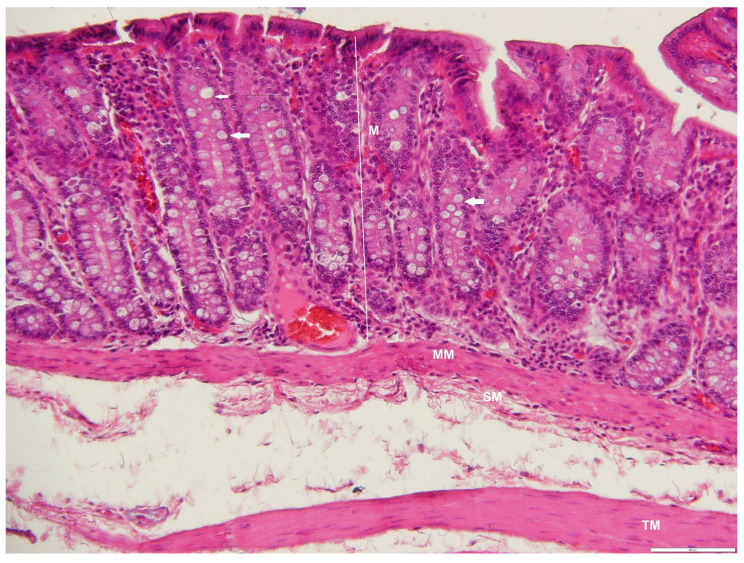
Photomicrograph of a cross-section through the rectum of an adult male *D. leporina*, showing goblet cells (arrows) H&Ex200. M—mucosa; MM—muscularis mucosa; SM—submucosa; TM—tunica muscularis.

**Table 1 animals-12-02493-t001:** The length (cm) and relative length (%) of various segments of the gastrointestinal tract of adult male *D. leporina.* (*n* = 6).

Parameter	Length (cm)	Relative Length (%)
Whole digestive tract	780.5 ± 50.62	_
Esophagus	16.41 ± 1.45	2.05
Stomach	12.70 ± 0.71	1.54
Small intestine	603.9 ± 10.15	77.42
Cecum	23.55 ± 0.64	3.02
Colon and rectum	123.55 ± 24.64	15.84

*Note:* Data are stated as mean ± standard deviation.

**Table 2 animals-12-02493-t002:** The histometric measurements (μm) of esophagus and stomach of adult male *D. leporina* (*n* = 6).

Parameters	Esophagus	Stomach		
		Cardiac	Fundic	Pyloric
T. mucosa	235.91 ± 19.12	421.71 ± 15.52	605.39 ± 28.68	523.26 ± 10.19
T. submucosa	465.67 ± 74.78	64.20 ± 4.14	77.26 ± 8.46	56.75 ± 16.24
T. muscularis	424.59 ± 11.12	274.66 ± 26.31	315.16 ± 23.14	439.91 ± 35.44
T. serosa	40.34 ± 9.49	13.51 ± 588	17.78 ± 7.43	12.08 ± 1.83
Total thickness	1023.78 ± 28.97	890.34 ± 22.76	978.78 ± 35.65	1034.87 ± 43.56

*Note:* Data are stated as mean ± standard deviation.

**Table 3 animals-12-02493-t003:** The histometric measurements (μm) of the small intestine of adult male *D. leporina* (*n* = 6).

Parameters	Small Intestine		
	Duodenum	Jejunum	Ileum
T. mucosa	213.00 ± 5.19	274.51 ± 7.80	232.85 ± 11.35
Villi Length	149.27 ± 22.97	182.50 ± 27.38	172.30 ± 12.51
Villi Width	43.10 ± 3.40	54.49 ± 3.31	35.66 ± 3.48
Crypts Length	22.65 ± 2.36	35.88 ± 3.85	27.40 ± 3.34
Crypts Width	21.13 ± 2.43	37.96 ± 3.96	26.84 ± 2.54
No. Goblet Cells/10,000 μm^2^	2.00 ± 0.63	5.50 ± 1.52	24.67 ± 2.73
T. submucosa	95.33 ± 3.33	109.90 ± 4.64	67.94 ± 6.13
T. muscularis	60.42 ± 7.99	141.94 ± 4.83	112.73 ± 4.55
T. serosa	8.70 ± 1.69	7.76 ± 2.70	7.33 ± 1.63
Total Thickness	345.23 ± 19.46	476.89 ± 23.78	405.24 ± 18.45

*Note:* Data are stated as mean ± standard deviation.

**Table 4 animals-12-02493-t004:** The histometric measurements (μm) of the large intestine of adult male *D. leporina* (*n* = 6).

Parameters	Large Intestine		
	Cecum	Colon	Rectum
T. mucosa	281.10 ± 8.89	241.96 ± 9.95	314.02 ± 8.76
No. Goblet Cells/10,000 μm^2^	26.00 ± 1.41	38.5 ± 4.32	24.50 ± 5.17
T. submucosa	56.62 ± 3.76	78.19 ± 1.85	121.48 ± 4.25
T. muscularis	328.97 ± 12.26	144.99 ± 34.91	218.36 ± 8.12
T. serosa	7.22 ± 1.17	9.01 ± 2.02	7.09 ± 2.46
Total thickness	680.98 ± 33.89	530.90 ± 22.87	630.54 ± 25.61

*Note:* Data are stated as mean ± standard deviation.

## Data Availability

All data was presented in the manuscript.

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
