# Peer review of "Morpho-Histological Studies of the Gastrointestinal Tract of the Orange-Rumped Agouti (Dasyprocta leporina Linnaeus, 1758), with Special Reference to Morphometry and Histometry"

_animals, 2022, doi:10.3390/ani12192493_

Round 1
Reviewer 1 Report
please see the pdf - yellow notes in the paper

Author Response
Based on the authors comments in text adjustments were made.
Information was provided on the diets in the discussion as well as future work on the nutritional requirements for the agouti.
Reviewer 2 Report
Major concern
A statistical analysis, not only descriptive, must be carried out, showing the differences between the structures studied, to validate the statements made in the discussion (e.g. differences in goblet cells between structures and villi in the different regions of small intestine in Table 3).
Some of the histological structures (e.g. crypts) are poor in appearance, so it would be better to add a few more photomicrographs at higher magnification to be able to differentiate the cellular structures.
In the panel of microphotographs (figure 3 and 4), the scale bar and references of each structure (e.g. MM, DG, etc...) and process (e.g. Lymphocyte infiltration) must be indicated with more intensity in the photomicrography.
Minor
Line 79-80. This sentence need more details about solutions used to conservation for further histologic examination.
Line 87. How did the distance calibration to measure each histologic structure? Did you use a slide with a micrometric scale for microscopy? please, added in the text.
Line 95 This sentence need more details about how many slides used for each animal for each structure.
Figure 1. b y a Photography not a photomicrography. A scale bar would be very helpful to visualize the sizes. The references of the structures (H, L, Lu, etc…) must be indicated with more intensity in the photography.
Line 108, 119, 128 and 129. please add standard deviation.
Figure 2. Change Small Intestines for Small Intestine.
Figure 4. H&E20. I guess it's H&E200?
Line 235 To evidence this sentence, show the esophageal glands in the photomicrograph of the esophagus.
Line 258. Why do the authors make this assumption? This reference does not support this affirmation.
Line 272 and 277. To validate these statements, statistical analysis must be performed to support this result.
Line 303. Add units.
Line 314. I suggest adding the study of digestive enzymes to improve the knowledge of the digestive tract.

Author Response
Major concern
A statistical analysis, not only descriptive, must be carried out, showing the differences between the structures studied, to validate the statements made in the discussion (e.g. differences in goblet cells between structures and villi in the different regions of small intestine in Table 3).
Authors response: I agree with the authors comments about the statistical work that must be carried out to make such statements in the discussion. However, the objective of this paper was to present novel descriptive work on the digestive system of an unknown species. As such revision were made in the discussion to highlight that there were numerical differences.
This work is really a pilot study which will initiate further work on the digestive system of this species were the effect of diet can be further studied using statistical tools.
Some of the histological structures (e.g. crypts) are poor in appearance, so it would be better to add a few more photomicrographs at higher magnification to be able to differentiate the cellular structures.
In the panel of microphotographs (figure 3 and 4), the scale bar and references of each structure (e.g. MM, DG, etc...) and process (e.g. Lymphocyte infiltration) must be indicated with more intensity in the photomicrography.
Authors' response: All photographs were adjusted and revised to address all issues raised by the reviewer.
Minor
Line 79-80. This sentence need more details about solutions used to conservation for further histologic examination.
Authors' Response: Further information was given on the solution and conservation of tissue in text. Thanks!
Line 87. How did the distance calibration to measure each histologic structure? Did you use a slide with a micrometric scale for microscopy? please, added in the text.
Authors' response: A micrometric scale was used for the microscopy. This information was included in text. Thanks!
Line 95 This sentence need more details about how many slides used for each animal for each structure.
Authors response: There number of slides per section was included in the manuscript (Three slides per section).
Figure 1. b y a Photography not a photomicrography. A scale bar would be very helpful to visualize the sizes. The references of the structures (H, L, Lu, etc…) must be indicated with more intensity in the photography.
Authors' comments: All photo micrographs were revised to addressed all issues raised by the reviewer.
Line 108, 119, 128 and 129. please add standard deviation.
Authors' comments: The standard deviations were included throughout the manuscript.
Figure 2. Change Small Intestines for Small Intestine.
Authors' response: This was adjusted.
Figure 4. H&E20. I guess it's H&E200?
Authors' comment: This was a typo, it was revised, Thanks.
Line 235 To evidence this sentence, show the esophageal glands in the photomicrograph of the esophagus.
Authors' response: There were no photo micrographic evidence presented, however, it was seen in some sections (very sparse and the phoptographs were not of good quality). As such the information in the text was changes represent this.
Line 258. Why do the authors make this assumption? This reference does not support this affirmation.
Authors' response: In the humble opinion of the authors the statement is supported by the work published. It showed an increase in the layers of the muscularis mucosa in the small intestines in comparison to the stomach and oesophagus. This increase in thickness can be due to the absorption of nutrients in the this section.
Line 272 and 277. To validate these statements, statistical analysis must be performed to support this result.
Authors' response: This study was done on a descriptive basis. Further work should be done on analytic data. Statements were made in text clarifying that this was numeric difference and not statistical difference.
Line 303. Add units.
Authors' response: These units were added.
Line 314. I suggest adding the study of digestive enzymes to improve the knowledge of the digestive tract.
Authors' response: These suggestions were added into the end of the manuscript. This comment was very valuable as it will direct future research.
Reviewer 3 Report
Manuscript entitled “Morpho-histological studies of the Gastrointestinal Tract of the Orange Rumped Agouti (Dasyprocta leporina Linnaeus, 1758): With Special Reference to Morphometry and Histometry” for the Special Issue “Advances in Wildlife and Exotic Animals Anatomy” of Animals.
The authors performed a comprehensive anatomical and histological study with special emphasis on morphometric and histometric measurements of the gastrointestinal tract of Agouti in order to understand the digestive biology of this animal.
The paper is clear and well detailed. The purpose of the study is appropriately defined. The details of the methods are comprehensible and consistent. The results are clearly presented. The discussion is relevant and complete. The manuscript contains sufficient and proper references. The Tables are correct. The figures are good. Therefore I would suggest accepting the manuscript for publication in Animals after minor revision.
a) Page 1, lines 18-19. “Based on the analysis of the gastrointestinal tract these animals can be fed a dry omnivorous diet with protein sources from either or animal matter”. This is not entirely clear. It should be rewritten.
b) GENERAL STRUCTURE OF THE DIGESTIVE TRACT (from Junqueira’s Basic Histology TEXT AND ATLAS. Anthony L. Mescher. 2018 by McGraw-Hill Education).
All regions of the GI tract have certain structural features in common. The GI tract is a hollow tube with a lumen of variable diameter and a wall made up of four main layers: the mucosa, submucosa, muscularis, and serosa.
â– The mucosa consists of an epithelial lining; an underlying lamina propria of loose connective tissue rich in blood vessels, lymphatics, lymphocytes, smooth muscle cells, and often containing small glands; and a thin layer of smooth muscle called the muscularis mucosae separating mucosa from submucosa and allowing local movements of the mucosa. The mucosa is also frequently called a mucous membrane.
â– The submucosa contains denser connective tissue with larger blood and lymph vessels and the submucosal (Meissner) plexus of autonomic nerves. It may also contain glands and significant lymphoid tissue.
c) Page 5, lines 138 to 143. “The mucosa of the oesophagus was folded and lined by stratified keratinized squamous epithelium (Figure 3a) with an average thickness of 1023.78 ± 28.97 μm. The lamina propria consisted of lymphocytic infiltration throughout the oesophagus. In addition, scattered lymph nodules were also found. The muscularis mucosae was distinct and consisted of smooth muscle fibres. The submucosa was the thickest layer at 465.67 ± 74.78 μm and consisted of a dense irregular connective tissue layer with abundant oesophageal glands, lymphocytes, blood vessels, and nerves”.
The main weaknesses of this manuscript is the description of the oesophagus.
The authors describe the presence of glands in the esophagus, but is this really the case? In what area of the oesophagus are they located? In the Figure 3a their absence is evident. The authors’ should be more precise with the identification of the oesophageal glands. It should be clarified.
They also confused the lamina propria of the mucosa with the submucosa in the Figure 3a. It should be corrected.
d) The Figures 3b and 3d are not clear enough. They should be replaced.
Author Response
Manuscript entitled “Morpho-histological studies of the Gastrointestinal Tract of the Orange Rumped Agouti (Dasyprocta leporina Linnaeus, 1758): With Special Reference to Morphometry and Histometry” for the Special Issue “Advances in Wildlife and Exotic Animals Anatomy” of Animals.
The authors performed a comprehensive anatomical and histological study with special emphasis on morphometric and histometric measurements of the gastrointestinal tract of Agouti in order to understand the digestive biology of this animal.
The paper is clear and well detailed. The purpose of the study is appropriately defined. The details of the methods are comprehensible and consistent. The results are clearly presented. The discussion is relevant and complete. The manuscript contains sufficient and proper references. The Tables are correct. The figures are good. Therefore I would suggest accepting the manuscript for publication in Animals after minor revision.
Authors response: Thank you for your favorable comments on the paper. All minor issues were corrected in text. We do hope to do further work to investigate the effect of various diets on the digestive system of the agouti.
- a)Page 1, lines 18-19. “Based on the analysis of the gastrointestinal tract these animals can be fed a dry omnivorous diet with protein sources from either or animal matter”. This is not entirely clear. It should be rewritten.
Authors' response: This was re written and the word 'vegetable' was included after 'either'.
- b)GENERAL STRUCTURE OF THE DIGESTIVE TRACT (from Junqueira’s Basic Histology TEXT AND ATLAS. Anthony L. Mescher. 2018 by McGraw-Hill Education).
All regions of the GI tract have certain structural features in common. The GI tract is a hollow tube with a lumen of variable diameter and a wall made up of four main layers: the mucosa, submucosa, muscularis, and serosa.
â– The mucosa consists of an epithelial lining; an underlying lamina propria of loose connective tissue rich in blood vessels, lymphatics, lymphocytes, smooth muscle cells, and often containing small glands; and a thin layer of smooth muscle called the muscularis mucosae separating mucosa from submucosa and allowing local movements of the mucosa. The mucosa is also frequently called a mucous membrane.
â– The submucosa contains denser connective tissue with larger blood and lymph vessels and the submucosal (Meissner) plexus of autonomic nerves. It may also contain glands and significant lymphoid tissue.
- c)Page 5, lines 138 to 143. “The mucosa of the oesophagus was folded and lined by stratified keratinized squamous epithelium (Figure 3a) with an average thickness of 1023.78 ± 28.97 μm. The lamina propria consisted of lymphocytic infiltration throughout the oesophagus. In addition, scattered lymph nodules were also found. The muscularis mucosae was distinct and consisted of smooth muscle fibres. The submucosa was the thickest layer at 465.67 ± 74.78 μm and consisted of a dense irregular connective tissue layer with abundant oesophageal glands, lymphocytes, blood vessels, and nerves”.
The main weaknesses of this manuscript is the description of the oesophagus.
The authors describe the presence of glands in the esophagus, but is this really the case? In what area of the oesophagus are they located? In the Figure 3a their absence is evident. The authors’ should be more precise with the identification of the oesophageal glands. It should be clarified.
Authors response: This was revised. The authors did find oesophageal glands, however, they were very scant and the photographs were not suitable for publication. Revisions were made, stating that oesophageal gland were scant and not numerous as mentioned before. Thank you for your comments
They also confused the lamina propria of the mucosa with the submucosa in the Figure 3a. It should be corrected.
Authors' comments: Thank you for noting this and revisions were made to the manuscript.
- d)The Figures 3b and 3d are not clear enough. They should be replaced.
Authors response: All figures were revised in the document
Round 2
Reviewer 2 Report
The main objective of this study, as the authors state, is to describe the digestive system of an unknown species. Nonetheless, I suggest that a sentence like “A more rigorous study, with controlled diet would be a better model to analyze in deep the histological microstructure” be added after the objective.
Figure 2 is a photograph not photomicrograph.
In the original MS appears the sentence: “A large number of oesophageal glands indicated that D. leporina can process roughage very well”. In the new version “A scant number of oesophageal glands (not shown) indicated that D. leporina can process roughage very well”. However, in the abstract (line 25), in results (line 163) and in conclusion (line 363) the authors affirm abundance of esophageal glands. Please, clarify this issue. In this regard, do Sukon, P.; Timm, K.I.; Valentine, B.A. (2009) specifically support the evidence about esophageal glands and fiber processing?
Author Response
The main objective of this study, as the authors state, is to describe the digestive system of an unknown species. Nonetheless, I suggest that a sentence like “A more rigorous study, with controlled diet would be a better model to analyze in deep the histological microstructure” be added after the objective.
Response: Thank you for your comments. A sentence addressing this issue was added to the objective. Future work with this animal will explore the effect of various diets on the gastrointestinal tract morphology, function, growth and carcass parameters.
Figure 2 is a photograph not photomicrograph.
Response: This was revised to photograph in the manuscript.
In the original MS appears the sentence: “A large number of oesophageal glands indicated that D. leporina can process roughage very well”. In the new version “A scant number of oesophageal glands (not shown) indicated that D. leporina can process roughage very well”. However, in the abstract (line 25), in results (line 163) and in conclusion (line 363) the authors affirm abundance of esophageal glands. Please, clarify this issue. In this regard, do Sukon, P.; Timm, K.I.; Valentine, B.A. (2009) specifically support the evidence about esophageal glands and fiber processing?
Response: Apologies for not being consistent throughout the manuscript. This was addressed and revised. The scant oesophageal gland give the agouti the ability to process some fibre.